# OpenReview forum: "AttnGCG: Enhancing Adversarial Attacks on Language Models with Attention Manipulation"
_NeurIPS.cc/2024/Conference — Submitted to NeurIPS 2024_

### Official Review · Reviewer_48Wr · 2024-07-08

**Soundness:** 3
**Presentation:** 3
**Contribution:** 2
**Rating:** 5
**Confidence:** 4

**Summary:**

The main claim of this paper is that adversarial suffixes against large language models (LLMs) function by distracting the model from the original harmful goal to the suffix itself. The authors then propose a modification to GCG attack by incorporating a regularization term that increases the attention score on the adversarial suffix.

**Strengths:**

### 1. Originality and significance

The main claim of the paper is an interesting hypothesis that aims to unfold the inner workings of adversarial attacks on LLMs. This type of question can lead to a nice interpretability tool and/or a potential mitigation. Hence, the significance of this research question is clear.

While some existing works start to look into “features” or neurons that correspond to these jailbreak attacks, the attention weights have not been deeply studied to the extent of my knowledge so it could be a nice complementary explanation.

### 2. Experiment coverage

The experiments on the attacks are relatively thorough. The authors compare their method against three existing SOTA attacks (GCG, AutoDAN, and ICA) on various open-source models. The transfer attack experiments in Section 3.4 also cover a broad range of closed-source models. The evaluation metrics are also comprehensive, including both the keyword matching and the GPT-4 evaluation.

**Weaknesses:**

### 1. Attention score measurement and interpretation

I first notice that in Figure 2, the attention scores on all parts (system, goal, suffix) can all go up as the optimization progresses and that the attention scores can be larger than 1 in Table 2 and 3. This suggests that the attention scores are before softmax and hence, not normalized to sum to 1. Please feel free to correct me if I’m mistaken.

1. If this is the case, it makes the score much more difficult to interpret and compare across different attacks. The absolute unnormalized value of the attention scores does not mean much because, for example, even if the score increases for the suffix portion, it may gets smaller relative to the other portions (system or goal). This is major flaw that undermines the main conclusion of the paper.
2. If that authors have not already done so, I would like to ask that all the reported attention scores be normalized (after softmax). The autoregressive generation also contributes to the attention scores, i.e., attention score of the target token $x_{n+2}$ also includes the target token $x_{t+1}$ along with all the prompt tokens $x_{1:n}$. I’m not sure what is the best way to normalize their effect. One way is to simply leave them out of the softmax, but there could be an interesting trend that we fail to capture this way. Another way is to report *difference* between average unnormalized attention score on the goal vs on the suffix portions. This also gives us a relative score but ignores the system portion.
3. In Figure 2 (left), ASR also increases along with the attention score on the goal, contradicting the main claim of the paper that higher attention score on the suffix is better.
4. It is unclear to me how Figure 5 supports the main claim of the paper. The attention pattern on "Vanilla" is strikingly similar to that of  "ICA" on the goal segment. Based on the color bar, the ICA attention score also seems higher than the Vanilla which contradicts the claim that the attack "diverts the model’s attention away from the goal towards themselves.”

### 2. Section 3.3: Generalize AttnGCG to other attack methods

1. The purpose of this experiment is unclear to me. If the authors wish to prove their claim that higher attention weight on the suffix leads to a better attack, there should be a better controlled experiments than running GCG or AttnGCG on prompts generated by the other methods. This experiment entangles the initialization method with the attention score.
2. It might be interesting to see AttnGCG with varying values of $w_t$ and $w_a$.
3. I’d suggest an experiment where the attention loss is incorporated into AutoDAN (or other attacks) optimization objective. This would better emphasize the transferability and the usefulness of the attention loss across multiple attack algorithms.

### 3. Limited empirical improvement

While the main idea could help improve interpretability to these adversarial attacks, the attack that is inspired by this observation, AttnGCG, does not lead to significant improvement in the attack success rate, especially in the transfer setting. In the white-box setting, the improvement seems consistent across models, but the small margin suggests that attention score is not the most important factor that determines the success of the attack.

That said, it is sufficiently convincing to me that AttnGCG performs better than GCG and may replace it for evaluating the safety of LLMs.

**Questions:**

1. In Figure 4 (right) and 5 (ICA, top left), why is most attention weight *concentrated on the first suffix token*? Does this mean that the first suffix token has the highest influence on the output?
2. In Figure 4, I would like to see the attention map for more samples and also for successful GCG and failed AttnGCG. There seems to be an interesting pattern here that is not captured by just the average attention score (reported in Table 2 and 3). Also, is the observation dependent on the optimization steps? Would it be possible to also visualize the attack at various steps other than 60?
3. Are the scores in Table 2 and 3 also average across the 100 samples?

**Limitations:**

Limitations and negative societal impact have been adequately addressed.

---

> ### Author Rebuttal · Authors · 2024-08-07
>
> We first thank the reviewer for the detailed comments and the appreciation of our work. We address the concerns below:
>
> $\textbf{Q1: Are attention scores normalized?}$
>
> Yes. The attention weights matrix is the value after softmax, and the attention weights of each target token are normalized to sum to 1. The reason why the attention score is greater than 1 in Tables 2 and 3 is that the attention score is not averaged over the target token length but added. For Figures 4 and 5, the goal and target used in this paper are the same (see Appendix B.3), so the values in Tables 2 and 3 are comparable.
>
> The reason that "in Figure 2, the attention scores of all parts increase as the optimization proceeds" is caused by the auto-regressive generation and the separator token (e.g., [INST]) in the model input. The decrease in attention to the auto-regressive generation and separator will lead to your observation.
>
> $\textbf{Q2: Proposals for alternative attention score normalization}$
>
> Thank you for your suggestions. We only consider the normalized attention weights (after softmax) of the goal and suffix in the model input, ignoring the separator. As for the attention weight matrix changing brought by the autoregressive generation, it is an inherent feature in the self-attention mechanism, which is not in the scope of this paper. And we make sure that this setting is consistent throughout the paper to make fair comparisons.
>
> $\textbf{Q3: Does Figure 2 (left) contradict the main claim of the paper?}$
>
> No, Figure 2 (left) does not contradict our main claim. First, a higher goal attention score does not necessarily mean a lower suffix attention score (See Q1 above). The attention score on the suffix also increases in Figure 2 (left), which supports our claim. Second, in Figure 2, our claim is mainly supported by the comparison between the left and right sides of the picture, not just from one side. Specifically, in both sides of Figure 2, the attention score on the goal converges to a certain low level (after 100 steps), which indicates a possibly successful attack. On the right side of Figure 2, a higher attention score on the suffix increases the probability of a successful attack in the same training steps compared with Figure 2 (left), which means increasing the suffix attention score can produce a more effective one for attack.
>
> $\textbf{Q4: The consistency between Figure 5 and the main claim of the paper}$
>
> Sorry for the confusion in the figure. We place the color bar and its concrete statistics aside from the figure. The ICA's goal attention score is lower than Vanilla based on the color bar. In Figure S.1 and Figure S.2 of the rebuttal supplementary material, we have added a clearer comparison and unified the color bar scale.
>
> $\textbf{Q5: Purpose of "Section 3.3: Generalize AttnGCG to other attack methods"}$
>
> The purpose of 3.3 experiment is to prove that AttnGCG is an attack method that can be easily integrated into other jailbreaks, which can further optimize the existing methods that have been optimized to convergence at the prompt level. We can seamlessly incorporate AttnGCG into other jailbreaking attacks through the initialization.
>
> $\textbf{Q6: Ablation experiment on $w_{t}$ and $w_{a}$}$
>
> Table S.2: Ablation for $w_{a}/w_{t}$ on Gemma-7b-it. The result format is "GPT-4 judge (keyword-detection)".
> | $w_{a}:w_{t}$ | 50 : 1              | 75 : 1              | 100 : 1             | 125 : 1             | 150 : 1             | 200 : 1             |
> |---------------|-----------------|-----------------|-----------------|-----------------|-----------------|-----------------|
> | Gemma-7b-it   | 64.0\% (89.0\%) | 68.0\% (90.0\%) | 75.0\% (92.0\%) | 73.0\% (91.0\%) | 72.0\% (89.0\%) | 67.0\% (90.0\%) |
>
> This table is also presented at Table S.2 in the rebuttal supplementary material.
>
> $\textbf{Q7: Additional experiments on other attack methods}$
>
> Thank you for your suggestions. We will add additional results in the next revision.
>
> $\textbf{Q8: Limited empirical improvement}$
>
> Thank you for your recognition. AttnGCG demonstrates consistent performance enhancements across diverse LLMs, with an average improvement of 7\% in the Llama-2 series and 10\% in the Gemma series. Possible applications of AttnGCG include replacing GCG in the future, as you have mentioned.
>
> $\textbf{Q9: Does the first suffix token have the highest influence on the output?}$
>
> No. It is a coincidence that the highest attention score is at the first suffix token in Figure 4(right). We will add more qualitative examples to showcase different successful attacking cases in the revision. In Figure 5 (ICA, top left), the attention weight is not concentrated on the first suffix token. Please refer to the actual input prompt of ICA in Appendix B.3 for token position matching.
>
> $\textbf{Q10: Attention maps for successful GCG and failed AttnGCG}$
>
> We show the attention maps of successful GCG and failed AttnGCG cases in Figure S.3 of the rebuttal supplementary material.
>
> $\textbf{Q11: Is the observation dependent on the optimization steps?}$
>
> No. For the optimization step, we choose the number that both AttnGCG and GCG start to converge, which is around the 60th step.
>
> $\textbf{Q12: Are the scores in Table 2 and 3 also average across the 100 samples?}$
>
> No. Tables 2 and 3 are the average values of the matrix in Figures 4 and 5. We use Tables 2 and 3 as the numerical explanations of these two visualizations.

---

> > ### Author Response · Authors · 2024-08-11
> >
> > Thanks for your time and comments on our work.
> >
> > In the rebuttal period, we provided detailed responses to all your comments and questions point-by-point regarding the unclear presentations. Specifically, we provided detailed explanations on
> >
> > Q1,2: About attention score normalization
> >
> > Q3,4: The consistency between Figure 2,5 and the main claim of the paper
> >
> > Q5: Purpose of experiments in Section 3.3
> >
> > Q6: Ablation experiment on $w_{t}$ and $w_{a}$
> >
> > Q9,10,11,12: Additional details in Figures and Tables
> >
> > Would you mind checking our responses and confirming whether you have any further questions?
> >
> > Any comments and discussions are welcome!
> >
> > Thanks for your attention and best regards.

---

> > > ### Comment · Reviewer_48Wr · 2024-08-11
> > >
> > > Thank you for all the clarifications. My main concern has been addressed. I am more convinced by the technical goal and how this paper achieves it. Please consider improving clarity of the paper and presentation of the results accordingly.
> > >
> > > The main contribution of the paper is still limited in my opinion, but I find the benefit of this paper outweighs its shortcomings. As a result, I decide to raise my rating to 5.

---

> > > > ### Author Response · Authors · 2024-08-12
> > > >
> > > > Thanks again for all your invaluable suggestions, we will revise the paper following the suggestions.
> > > >
> > > > And we are glad to see your concerns have been addressed! Thanks for raising the score!

---

### Official Review · Reviewer_yXuJ · 2024-07-08

**Soundness:** 3
**Presentation:** 3
**Contribution:** 2
**Rating:** 4
**Confidence:** 3

**Summary:**

This work proposes a new adversarial attack strategy on LLMs which improves over existing adversarial attacks. For this the authors propose a new regularizer that maximizes the weight of attention corresponding to suffix tokens, which naturally results in minimizing the weight for the other tokens present in the input prompt. Using this additional regularizer with GCG results in improved attack success rate. The authors also show that this attack is transferable to other attack methods like ICA and AutoDAN.

**Strengths:**

1) The paper is well motivated and the proposed loss follows well with the reader’s intuition.
2) The results are promising and the gains over the existing GCG attack are significant.
3) The comparison is comprehensive, involving different models.

**Weaknesses:**

1) It is not clear how the transferability of the same suffix tokens is for different goal prompts. This is important to investigate because GCG shows that the generated attacks are universal and can transfer on different goal prompts. I am currently a bit skeptical that the transferability on using the proposed attack might be limited because the generated suffix tokens might be more specialized for the given goal prompt. This is expected because now the generation of the suffix tokens is largely conditioned on the target target tokens due to the proposed regularizer.

2) I believe it might be possible that using the proposed attack the model ends up outputting something potentially harmful but completely unrelated with the input prompt. This might be a possibility because the proposed approach inherently minimizes the attention on the goal tokens, which means the context of the input might become less relevant. It would be great if the authors could share some analysis on transferability of adv prompts and also share the generated text for GCG and AttnGCG.

3) It is not clear why maximizing the attention weights for suffix tokens should always lead to a stronger attack? This is also evident from tables 2 and 3 where AutoDAN has a lower value of goal attention score but stil leads to weaker attack as compared to GCG (see Table-4). Thus the argument presented in 162-163 seems questionable.
In general, it is not clear why authors did not attempt to analyze the defenses like the ones proposed in [1]. Particularly, I believe it is important to analyze if the proposed attacks are able to bypass detection filters based on perplexity [1].

[1] Jain, Neel et al. “Baseline Defenses for Adversarial Attacks Against Aligned Language Models.” ArXiv abs/2309.00614 (2023)

**Questions:**

I request the authors to kindly address the questions in the weaknesses section.

**Limitations:**

Yes, the authors have addressed the limitations.

---

> ### Author Rebuttal · Authors · 2024-08-07
>
> We first thank the reviewer for the detailed comments and the appreciation of our work. We address the concerns below:
>
> $\textbf{Q1: Concerns about transferability of adv prompts across goals}$
>
> Thank you for your suggestion of adding transfer experiments across different goal prompts. We will conduct a "multiple-behavior" experiment in [1] in the next version.
>
> $\textbf{Q2: Concerns about possibly generating a harmful but unrelated response with the input prompt}$
>
> Thank you for bringing the question up. Our GPT-4 metric does not count "harmful but completely unrelated with the input prompt" cases into the ASR result. The GPT4-judge determines whether the model answers the input request accurately. Therefore, the ASR from GPT4-judge refers to harmful answers in the interests of the attacker (see the "Prompt template for GPT-4 judge" in Appendix A.1). The experimental results show that the ASR of AttnGCG has increased (see Table 1), indicating LLMs poisoned by AttnGCG answers harmful questions more correctly than LLMs attacked by other methods. We also show generated responses of GCG and AttnGCG in Appendix B.3.
>
> $\textbf{Q3: The relationship between maximizing the attention weights for suffix and a stronger attack}$
>
> Maximizing the attention weight of the suffix token aims to improve the LLM jailbreak, making the model more inclined to answer the user's request. However, the "Maximization" of suffix attention has a boundary, that is, the answer content must be related to the goal.
> In AttnGCG, we use Target Loss $L_{t}$ as the regulator. As presented in the experiments in section 3.2, AttnGCG has a better attack effect, which verifies the claim.
>
> The reason why "AutoDAN leads to a weaker attack with a lower goal attention score" is that a low goal attention score will steer a model's response to be irrelevant with the goal. Please note that, our AttnGCG is not designed to minimize the goal attention in the input, which is also clarified in the first point replied to reviewer Hgf8.
>
> $\textbf{Q4: Concerns about capability to bypass perplexity-based defense}$
>
> Our AttnGCG, along with other jailbreaking methods that generate adversarial suffixes, are unlikely to bypass the perplexity-based defenses. This is because the adversarial suffix always has a higher PPL than natural language, which can be detected via the PPL metric easily.
> However the perplexity-based defense is a deployment-level method --- it can be deployed before instead of during using an LLM. The main contribution of AttnGCG lies in the methodology aspect. (1) In the future, we can try to use other deployment-level methods to bypass the perplexity-based defense. (2) The new optimization objective of attention score can also assist other methods in the future (e.g. Attn-AutoDAN). AttnGCG is a heuristic verification.
>
> [1] Universal and Transferable Adversarial Attacks on Aligned Language Models

---

> > ### Author Response · Authors · 2024-08-11
> >
> > Thanks for your time and comments on our work.
> >
> > In the rebuttal period, we provided detailed responses to all your comments and questions point-by-point regarding the unclear presentations. Specifically, we provided detailed explanations on
> >
> > Q1: Concerns about transferability across goals
> >
> > Q2: Concerns about possibly generating unrelated responses
> >
> > Q3: The relationship between attention weights for suffixes and attack performance
> >
> > Q4: Concerns about bypassing perplexity-based defense
> >
> > Would you mind checking our responses and confirming whether you have any further questions?
> >
> > Any comments and discussions are welcome!
> >
> > Thanks for your attention and best regards.

---

> > ### Comment · Reviewer_yXuJ · 2024-08-12
> >
> > Thank you for your efforts in the rebuttal. I think my questions are not addressed adequately in the current version. For instance, the use of adversarial attacks like GCG remains unclear if a metric as simple as perplexity can detect these samples. Although I understand the motivation for authors was to improve over GCG. But I think more discussion is required on this in the paper. Authors should perhaps provide rigorous evaluations on pre-processing based defenses as well.
> >
> > My second concern regarding generation of harmful but unrelated responses is also not adequately resolved. I would encourage the authors to provide a fine-grained analysis on how the text quality changes with increase in strength of the attack. Authors can use perplexity as a metric for this analysis. Additionally, they can perform some analysis using another LLM asking it to tell if the output generated correlates with the input provided.
> >
> > Overall I think this is an interesting paper, but perhaps might need some refinement and more detailed analysis. Therefore, I prefer to keep my score.

---

> > > ### Author Response · Authors · 2024-08-12
> > >
> > > Thank you for your suggestions, the PPL metric is a promising way to further evaluate LLM attack method, and we will incorporate this analysis in the final version.
> > >
> > > As for a more fine-grained evaluation that 'using another LLM asking it to tell if the output generated correlates with the input provided', this is one of the criteria we employed for GPT4 evaluation --- GPT4-judge determines whether the model answers the input request accurately. That is to say, our metric GPT4-judge only considers **harmful** and **accurate** (which is input-correlated) model response into a successful jailbreak.
> > >
> > > Also, please note that a higher suffix attention score does not necessarily mean a lower goal attention score (See Q1 replied to reviewer 48Wr). Our AttnGCG is not designed to "minimize" the goal attention. The optimization objective contains Attention Loss $L_{a}$ and Target Loss $L_{t}$. Target Loss $L_{t}$ ensures that the response will focus on the original Target content, thereby limiting the goal's attention score from being too low. Figure 2 (right) in the paper can support this view. It can be observed that the attention score on the goal converges approximately after step 100.

---

> > > ### Author Response · Authors · 2024-08-13
> > >
> > > We appreciate your feedback once again and hope that we have addressed all your concerns. Is there anything else you would like us to address?

---

### Official Review · Reviewer_CeR5 · 2024-07-09

**Soundness:** 2
**Presentation:** 3
**Contribution:** 2
**Rating:** 5
**Confidence:** 4

**Summary:**

The authors propose a refined GCG method named AttnGCG for Large Language Model jailbreaking attacks. They focus on the attention scores of the input components, refining the loss function by adding an Attention Loss term. The attack success rates are greatly improved. Various experiments are provided to support the effectiveness of the proposed method.

**Strengths:**

(1)	An interesting finding is that as the attention score on the adversarial suffix increases, the effectiveness in safeguarding LLM diminishes.

(2)	The experiments are conducted on various LLMs to prove the effectiveness of the AttnGCG.

**Weaknesses:**

(1)	It is unclear whether the increased success cases correspond to the 'regret' cases observed in GCG. The authors proposed AttnGCG to address the issue where the model successfully generates target tokens but then rejects the request; however, the results remain ambiguous.

(2)	In the success case illustrated in Figure 4, the attention scores at the boundary between the goal and the suffix are significantly higher than in other regions. Is this a common phenomenon in success cases? If so, why does this occur?

(3)	In Appendix A.3, the table shows that the system prompt for Llama-2 and Llama-3 is set to None, which is different from most jailbreaking papers, including the original GCG. How does this influence the attacking success rate? The authors should also report the success rate under the standard system prompt.

I will reconsider my score if all these problems are adequately addressed.

**Questions:**

GCG-based methods can be easily defended by perplexity-based defenses [1]. Can the proposed AttnGCG reduce the perplexity of the suffix? Or is there any insight to bypass such defenses further?

[1] Baseline Defenses for Adversarial Attacks Against Aligned Language Models

**Limitations:**

The authors adequately discussed the limitations of this work.

---

> ### Author Rebuttal · Authors · 2024-08-07
>
> We first thank the reviewer for the detailed comments and the appreciation of our work. We address the concerns below:
>
> $\textbf{Q1: Ambiguous results about solving failed 'regret' jailbreaking cases in GCG.}$
>
> The 'regret' jailbreaking case mentioned is a subcase of failed jailbreakings, which is caused by "a high probability of harmful tokens does not necessarily equate to a successful jailbreak" (L36). And our GPT-4 evaluator takes such failed jailbreaking case into account. In detail, the GPT-4 judge will only consider a case to be a successful attack if and only if the model respond to the request accurately.
> That is to say, the 'regret' situation will be considered a failed attack. In the experiments, the GPT-4 evaluated ASR of our method is improved (Table 1), demonstrating a better capacity of AttnGCG to handle this scenario.
>
> $\textbf{Q2: Is the phenomenon about the position of high attention score in Figure 4 common in success cases?}$
>
> No, this is not a common phenomenon in successful cases. We will add more qualitative examples to showcase different successful attacking cases in the revision.
>
> $\textbf{Q3: Concerns about system prompt settings for Llama series.}$
>
> Thank you for raising this question. Unlike Llama2, Llama3 was released without a specified system prompt. With the chat template of Llama3 changed significantly compared with Llama2, we did not use the official system prompt of Llama2 for Llama3, instead, we set it to None. For a fair comparison across the Llama series, we then set the system prompt of Llama2 to None. And also note that, the system prompt of Llama2 in AutoDAN [1] is also set to None, which is of reference value.
>
> The ASR results of Llama2-7b-chat with its official system prompt are reported below (also in Table S.1 in the rebuttal supplementary material). We can observe that the Llama2 with its official system prompt is more difficult to breach, requiring more steps to converge. We will add results of Llama3 in the revision.
>
> Table S.1: Results of Llama2-7b-chat after enabling the standard system prompt (the criterion for stopping optimization is Loss convergence, which is 1000 steps in the experiment, and the other parameters are the same). The data format is "GPT-4 judge (keyword-detection)".
> | Models          | GCG             | AttnGCG        |
> |-----------------|-----------------|----------------|
> | Llama-2-Chat-7B | 46.0\% (51.0\%) | 57.0\% (57.0\%) |
>
> $\textbf{Q4: Concerns about the capability to bypass perplexity-based defense.}$
>
> Our AttnGCG, along with other jailbreaking methods that generate adversarial suffixes, are unlikely to bypass the perplexity-based defenses. This is because the adversarial suffix always has a higher PPL than natural language, which can be detected via the PPL metric easily.
> However the perplexity-based defense is a deployment-level method --- it can be deployed before instead of during using an LLM. The main contribution of AttnGCG lies in the methodology aspect. (1) In the future, we can try to use other deployment-level methods to bypass the perplexity-based defense. (2) The new optimization objective of attention score can also assist other methods in the future (e.g. Attn-AutoDAN). AttnGCG is a heuristic verification.
>
> [1] AutoDAN: Generating Stealthy Jailbreak Prompts on Aligned Large Language Models

---

> > ### Author Response · Authors · 2024-08-11
> >
> > Thanks for your time and comments on our work.
> >
> > In the rebuttal period, we provided detailed responses to all your comments and questions point-by-point regarding the unclear presentations. Specifically, we provided detailed explanations on
> >
> > Q1: Ambiguous results in solving 'regret' cases
> >
> > Q2: Is the phenomenon in Figure 4 common?
> >
> > Q3: Concerns about system prompt settings
> >
> > Q4: Concerns about bypassing perplexity-based defense
> >
> > Would you mind checking our responses and confirming whether you have any further questions?
> >
> > Any comments and discussions are welcome!
> >
> > Thanks for your attention and best regards.

---

> > ### Author Response · Authors · 2024-08-13
> >
> > Thank you in advance for your feedback. Please let us know if we have addressed your concerns.

---

> > > ### Comment · Reviewer_CeR5 · 2024-08-14
> > >
> > > Thank you for the rebuttal. I have three additional suggestions:
> > >
> > > 1. The authors should also report the results under appropriate system prompts, including Llama 3: “You are a helpful, respectful, and honest assistant. Always answer as helpfully as possible, while being safe. ···’
> > >
> > > 2. Except for releasing the code, the generated suffix should also be released to help the reproducibility.
> > >
> > > 3. An additional dataset, JailbreakBench [1] may be reported to enhance the conclusion of this work.
> > >
> > > [1] https://jailbreakbench.github.io/
> > >
> > > Overall, I will keep my rating to support a borderline acceptance.

---

> > > > ### Author Response · Authors · 2024-08-14
> > > >
> > > > Thank you for these additional valuable suggestions, which are important for furthering the quality of our work. We will incorporate these analyses in the next version.

---

### Official Review · Reviewer_Hgf8 · 2024-07-11

**Soundness:** 2
**Presentation:** 3
**Contribution:** 2
**Rating:** 5
**Confidence:** 3

**Summary:**

The paper proposes a new jailbreak attack method against LLMs, called AttnGCG. The method integrates a loss of maximizing the attention scores of the adversarial suffix. The paper provides experimental results to show the effectiveness of the proposed method.

**Strengths:**

- The paper is well-written and easy to follow.

**Weaknesses:**

My main concerns are as follows.

- Will increasing the attention scores of adversarial suffixes make the responses focus on the content in adversarial suffixes?
- The discussion in lines 151-164 is weak. Specifically, in Figure 4, AttnGCG explicitly increases the attention scores of adversarial suffixes, so it is natural to have higher adversarial suffix attention scores. It is not convincing to say "uncover the underlying reasons for successful attacks within the model’s attention mechanism".
- In Table 3, AutoDAN achieves 0.227 goal attention score, while the scores of GCG and AttnGCG are 0.8657 and 0.793. Does the observation mean that AutoDAN is better than AttnGCG?
- Some content seems to be redundant, e.g., Figure 1 and Algorithm 1.

**Questions:**

See weakness.

---

> ### Author Rebuttal · Authors · 2024-08-07
>
> We first thank the reviewer for the detailed comments and the appreciation of our work. We address the concerns below:
>
> $\textbf{Q1: Will increasing attention scores for adversarial suffixes prioritize their content in responses?} $
>
> No, increasing the attention scores of adversarial suffixes will not make the responses focus on the content in the adversarial suffixes. The optimization objective contains Attention Loss $L_{a}$ and Target Loss $L_{t}$. Target Loss $L_{t}$ ensures that the response will focus on the original Target content, thereby limiting the goal's attention score from being too low. Figure 2 (right) in the paper can support this view. It can be observed that the attention score on the goal converges approximately after step 100.
>
> $\textbf{Q2: Ambiguous discussions in the attention score visualization.} $
>
> Sorry for the confusion. We clarify that the purpose of visualizing the Attention Map is to visually verify the effectiveness of our method. Specifically, we expect Figure 4 can verify that "In the successful jailbreaking case, attention is notably shifted to the suffix part, resulting in a decrease in attention from the goal", that is, reducing the model’s excessive attention to the goal and thus "bypassing the internal safety protocol", and reducing the model’s excessive attention to the goal is achieved by increasing the attention score of the adversarial suffix. Experiments show that higher attention scores for adversarial suffixes mean more effective adversarial suffixes and higher ASR (Figure 2, Table 1 in the paper), which supports our claim.
>
> $\textbf{Q3: Is AutoDAN better than AttnGCG?} $
>
> No, AutoDan is not always better than our method. From Table 4, the attack effect of AutoDAN is worse than AttnGCG, although AutoDAN has a lower attention score on the goal. This is because the LLM may respond irrelevant content to the goal with lower attention score on the goal (that's why we need the target loss as a regulator). In Table 4, the ASR of AutoDAN on keyword-detection is similar to AttnGCG, indicating that LLMs attacked by either method do not refuse to answer requests, but the ASR of AutoDAN on GPT-4 judge is much lower because the answer generated by AutoDAN are not recognized as an accurate one to the input request.
>
> $\textbf{Q4: Redundant content.} $
>
> Thanks for the suggestion. We will keep Figure 1 as the teaser, aiming to visually compare the difference and improvement between AttnGCG and GCG; we will remove Algorithm 1.

---

> > ### Author Response · Authors · 2024-08-11
> >
> > Thanks for your time and comments on our work.
> >
> > In the rebuttal period, we provided detailed responses to all your comments and questions point-by-point regarding the unclear presentations. Specifically, we provided detailed explanations on
> >
> > Q1: How is the response relevant to the goal?
> >
> > Q2: Clarification of ambiguous discussions
> >
> > Q3: Is AutoDAN better than AttnGCG?
> >
> > Q4: About redundant content
> >
> > Would you mind checking our responses and confirming whether you have any further questions?
> >
> > Any comments and discussions are welcome!
> >
> > Thanks for your attention and best regards.

---

> > ### Author Response · Authors · 2024-08-13
> >
> > Thank you in advance for your feedback. Please let us know if we have addressed your concerns.

---

> > > ### Comment · Reviewer_Hgf8 · 2024-08-14
> > >
> > > Thank you for the detailed responses. I suggest reorganizing the analysis part, as it currently confuses the readers. Evaluations of how the attack success rate changes with the choice of attention weight should also be added to the paper. Overall, I still think the paper is on the borderline, and I will increase the score to 5.

---

> > > > ### Author Response · Authors · 2024-08-14
> > > >
> > > > Thank you for your suggestions. We will revise the paper to further improve its clarity.
> > > >
> > > > As for "Evaluations of how the attack success rate changes with the choice of attention weight", we did an ablation experiment on gemma-7b-it during the rebuttal period, and the results are as follows.
> > > >
> > > > Table S.2: Ablation for $w_{a}/w_{t}$ on Gemma-7b-it. The result format is "GPT-4 judge (keyword-detection)".
> > > > | $w_{a}:w_{t}$ | 50 : 1              | 75 : 1              | 100 : 1             | 125 : 1             | 150 : 1             | 200 : 1             |
> > > > |---------------|-----------------|-----------------|-----------------|-----------------|-----------------|-----------------|
> > > > | Gemma-7b-it   | 64.0\% (89.0\%) | 68.0\% (90.0\%) | 75.0\% (92.0\%) | 73.0\% (91.0\%) | 72.0\% (89.0\%) | 67.0\% (90.0\%) |
> > > >
> > > > This table is also presented in Table S.2 in the rebuttal supplementary material.
> > > >
> > > > We will add the complete ablation experiment to the paper in the next version. Thank you again for your suggestions and raising the score!

---

### Author Rebuttal · Authors · 2024-08-07

First, we thank all reviewers for their insightful comments. We are particularly encouraged that reviewers have appreciated:
- The novelty and impact of our central ideas: "... lead to a nice interpretability tool and/or a potential mitigation. Hence, the significance of this research question is clear"(48Wr), "While some existing works..., so it could be a nice complementary explanation"(48Wr).
- The benefits of our proposed method: "The results are promising and the gains over the existing GCG attack are significant"(yXuJ), "The attack success rates are greatly improved"(CeR5), "AttnGCG performs better than GCG and may replace it for evaluating the safety of LLMs"(48Wr).
- The thorough coverage of our experiments: "Various experiments are provided"(CeR5), "The comparison is comprehensive"(yXuJ), "The experiments on the attacks are relatively thorough"(48Wr), "The evaluation metrics are also comprehensive"(48Wr).
- The motivation of our work: "The paper is well motivated and the proposed loss follows well with the reader’s intuition"(yXuJ).
- The quality of the paper writing: "The paper is well-written and easy to follow"(Hgf8).

Individual concerns have been addressed carefully in the response to each reviewer. In the final version, we will revise the paper following the suggestions.

---

### Decision · Program_Chairs · 2024-09-25

**Decision:**

Reject

**Comment:**

This paper observed a position correlation between a successful jailbreak and a high attention score on the adversarial suffix. Based on it, this paper proposed AttnGCG for improving the effectiveness of jailbreak attacks. This paper is a borderline paper and  contains mixed scores. AC has discussed this paper with reviewers. The remaining concerns from the reviewers of this paper are 1) analysis to ensure that the generated response indeed relates with the input prompt; and 2)perplexity based safety filters. Overall, AC understood that the authors use the existing standard evaluation metrics to evaluate the performance. But AC agrees with the reviewer and thinks it is necessary to evaluate whether the output is related to the question ( although AC understood the authors still use a L_t to constrain the goal).). For ppl based safety filter, it is reasonable to discuss it.

AC feels the reviewers' concerns are reasonable, and the authors should provide evaluation to address the above concerns, and the current version is slightly not ready for NeurIPS.